# One Dose of a Novel Vaccine Containing Two Genotypes of Porcine Circovirus (PCV2a and PCV2b) and *Mycoplasma hyopneumoniae* Conferred a Duration of Immunity of 23 Weeks

**DOI:** 10.3390/vaccines9080834

**Published:** 2021-07-29

**Authors:** José Carlos Mancera Gracia, Megan Smutzer, Lucas Taylor, Mónica Balasch, Meggan Bandrick

**Affiliations:** 1Zoetis Belgium S.A., Mercuriusstraat 20, 1930 Zaventem, Belgium; 2Zoetis Inc., Veterinary Medicine Research and Development, 333 Portage St., Kalamazoo, MI 49007, USA; megan.smutzer@zoetis.com (M.S.); lucas.p.taylor@zoetis.com (L.T.); meggan.bandrick@zoetis.com (M.B.); 3Zoetis Manufacturing & Research Spain S.L., Ctra. Camprodon s/n, Finca La Riba, 17813 Vall de Bianya, Spain; monica.balasch@zoetis.com

**Keywords:** swine, porcine circovirus 2, *Mycoplasma hyopneumoniae*, vaccine, duration of immunity

## Abstract

Porcine circovirus type 2 (PCV2) and *Mycoplasma hyopneumoniae* (*Mhyo*) are important swine pathogens for which vaccination is a key control strategy. Three separate studies were performed to evaluate the duration of immunity (DOI) conferred by a novel vaccine combining PCV2a/PCV2b and *Mhyo* into a ready-to-use formulation. In each study, three-week-old naïve piglets were vaccinated (Day 0) and challenged 23-weeks later (Day 159) with either PCV2a, PCV2b or *Mhyo*. Pigs were euthanized three-to-four-weeks post-challenge. Vaccinated pigs had significantly lower PCV2 viremia from Day 168 until Day 175 (PCV2a study) or until euthanasia (PCV2b study), respectively. Fecal shedding was significantly lower for PCV2a-challenged from Day 171 until Day 178, and for PCV2b-challenged from Day 172 until euthanasia. In the PCV2a challenge study, there were no differences among vaccinates and controls in terms of percent of pigs positive for PCV2 immunohistochemistry, histiocytic replacement, or lymphoid depletion. However, significant differences for immunohistochemistry and histiocytic replacement, not lymphoid depletion, were observed among vaccinates and controls following PCV2b challenge. Vaccination supposed a significant reduction in the mean percentage of *Mhyo*-like lesions in the lung. Percentages of lung tissues positive for *Mhyo* via immunohistochemistry were 49.3% and 67.1% for vaccinated and control groups, respectively. One dose of the novel PCV2a/PCV2b/*Mhyo* vaccine conferred robust protection against challenge 23-weeks later for all three fractions.

## 1. Introduction

The Porcine Respiratory Disease Complex (PRDC) is a disease of swine caused by a combination of viral (porcine circovirus type 2 (PCV2), porcine reproductive and respiratory syndrome virus (PRRSV), pseudorabies virus (PRV), swine influenza virus (SIV)), bacterial (*Actinobacillus pleuropneumoniae, Bordetella bronchiseptica, Mycoplasma hyopneumoniae* (*Mhyo*), *Pasteurella multocida*, *…)* and adverse management conditions (overcrowding, poor ventilation, temperature, …). PRDC main clinical manifestations are coughing, dyspnea, poor growth, and increased mortality. This syndrome can result in significant economic loses to the swine industry due to medication expenses, increased mortality, and the failure to gain weight during the end of the fattening period [1]. PCV2 and *Mhyo* are accounted as two of the most significant pathogens involved in the PRDC.

PCV2 is a single-stranded nonenveloped DNA virus, which has been described as the etiological agent of post-weaning multi-systemic wasting disease syndrome (PMWS) in the 1990s. PCV2 was later related to other clinical manifestations such as the already mentioned Porcine Respiratory Disease Complex (PRDC) or the Porcine Dermatitis and Nephropathy Syndrome. All those clinical manifestations are known as PCV2-associated diseases (PCVAD) and mainly cause disease in pigs around 5 to 18 weeks old. The primary transmission route of PCV2 is by oronasal exposure and the virus has been isolated in most pig secretions such as nasal and ocular secretions, feces, saliva, or urine [2,3]. Due to their single stranded DNA nature and small genomic size, PCV2 viruses have a very high evolutionary rate. In fact, PCV2 viruses have a mutation frequency of around 10^3^–10^4^ substitutions/site/year, which is one of the highest rates among DNA viruses and it is at the level of the variation seen in RNA viruses [4]. This high mutation rate, together with the high rate of recombination (second to PCV2′s propensity for multi-strain or multi-genotype co-infections and infection persistence), immune pressure lead to the genesis of novel and divergent PCV2 variants. These various mechanisms of evolution explain that until today up to eight different genotypes, including two recombinant ones have been described (PCV2a-h) [5,6]. Those genotypes were proposed based a maximum intra-genotype *p*-distance 13% (based on the ORF2 gene, which encodes the capsid protein) [6]. This information gives an idea on the complexity and the divergent scenario posed by PCV2 virus. However, although eight different genotypes were described nowadays there is a co-existence of three main PCV2 genotypes which are persistently causing disease in the field [7]. The first genotype identified was PCV2a, which was dominant until the early 2000s [8]. Since then, PCV2b was isolated and became predominant in commercial swine populations; PCV2b was related with more severe PCVAD clinical manifestations [9,10,11,12,13]. Finally, in 2011 PCV2d emerged as an evolution of PCV2b and took over as the predominant genotype worldwide [14,15]. *Mhyo* causes the enzootic pneumonia in pigs. Enzootic pneumonia is a respiratory syndrome that poses a great economic burden to swine production all around the world. *Mhyo* primarily infects ciliated epithelial cells on the surface of bronchioles, bronchi, and trachea causing cilia and epithelial cell damage. Although *Mhyo* affects pigs of all ages, clinical signs are mostly apparent in growing and finishing pigs. As a major contributor to PRDC *Mhyo* is routinely identified as concomitant agent with other infectious agents such as PRRSV or PCV2. As a strict respiratory pathogen *Mhyo* can be found in nasal secretions of diseased pigs. Therefore, transmission occurs via direct, nose-to-nose, contact [16,17]. In consequence, due to the threat posed by both pathogens (PCV2 and *Mhyo*) their prevention and control are crucial steps to successfully manage PRDC, even when other infectious and non-infectious factors are present [18].

Vaccination is one of the principal strategies to decrease the impact of PCV2 and *Mhyo* in the field. PCV2 vaccines are usually applied to piglets as a single dose around week 3 of age and aim to prevent PCV2 effects post weaning. Despite circulation of different PCV2 genotypes, most commercial vaccines worldwide still target PCV2a only [19,20]. Although PCV2a vaccines generate neutralizing antibodies that cross-react and are efficacious in avoiding clinical disease produced by heterologous strains, there are concerns about their ability to prevent viral replication or transmission [21,22]. As a consequence, there is an increasing deliberation around update of PCV2 vaccines in order to increase vaccine efficacy [20]. *Mhyo* commercial vaccines are mostly applied to piglets as a single dose between the first and the third weeks of age and aim to prevent the effects of *Mhyo* occurring after weaning [23]. As a consequence, due to the parallelisms seen on both infections and that vaccination of both pathogens takes place in piglets around the same age, combined vaccination has been explored since 2010 [24].

One of the variables that characterizes the efficacy of a vaccine is the duration of immunity (DOI). The DOI of a vaccine determines at least how long after vaccination a vaccine can confer robust protection. With the aim of assessing the DOI conferred by one dose of a novel vaccine combining PCV2a/PCV2b and *Mhyo*, three separate studies were performed. In each independent study three-week-old piglets naïve to PCV2 or *Mhyo* were vaccinated once and challenged 23 weeks later with either PCV2a, PCV2b, or *Mhyo.* Inmunological, virological/bacterial, and clinical variables were considered to measure the protective efficacy of the vaccine after the experimental challenge. 

## 2. Materials and Methods

### 2.1. Study Design

One independent study was completed to evaluate the duration of immunity generated by each of the three different components of the vaccine: PCV2 genotype a, PCV2 genotype b, and *Mhyo*. All independent studies and groups present in this manuscript are summarized in Table 1.

Study 1 evaluated the duration of immunity of the PCV2a fraction of the vaccine. Sixty-one piglets seronegative for PCV2 antibodies (by indirect ELISA) and without detectable PCV2 DNA (RT-qPCR) in blood were selected. Thirty pigs were randomly allotted to the control group vaccinated only with the *Mhyo* fraction of the vaccine (T01). The remaining thirty-one animals were enrolled in the vaccinated group, vaccinated with the complete vaccine containing the PCV2a, PCV2b, and *Mhyo* fractions (T02). Study 2 evaluated the duration of immunity of the PCV2b fraction of the vaccine. Sixty piglets also seronegative for PCV2 antibodies and without detectable PCV2 DNA in blood were selected. Thirty were randomly allotted to the control group vaccinated with the *Mhyo* fraction only (T03) and the remaining thirty were enrolled to the vaccinated group, containing the complete vaccine (T04). Finally, study 3 aimed to evaluate the duration of immunity of the vaccine’s *Mhyo* fraction. One hundred and fifty-two piglets seronegative for *Mhyo* antibodies were selected. Ten piglets were randomly allotted to the non-vaccinated control group (T05) and were used to evaluate the health of the animals prior to *Mhyo* challenge; seventy-one animals were allotted to the vaccine control group and were vaccinated only with the PCV2a and PCV2b fractions (T06). Finally, the remaining seventy-one pigs were enrolled to the vaccinated group and were vaccinated with all three fractions (T07).

Pigs were vaccinated around the third week of age with the complete vaccine (T02, T04, and T07). Animals from PCV2 control groups (T01 and T03) received a 2 mL dose of the mentioned knockout placebo containing only *Mhyo* bacterin via the intramuscular route. On the other hand, animals from *Mhyo* control group (T06) received 2 mL intramuscular dose of the mentioned knockout placebo containing PCV2a and PCV2b fractions, animals from the non-vaccinated control group (T05) did not receive any vaccine product. Twenty-three weeks after vaccination all animals (with the exception of T05) were challenged with the correspondent pathogen as specified in the challenge section. Euthanasia and necropsy occurred 3 to 4 weeks after challenge. In studies 1 and 2, serum was collected for evaluation of PCV2-specific antibodies and PCV2 viremia. Fecal swabs were collected for evaluation of PCV2 shedding. At necropsy lymphoid tissues (inguinal/mesenteric/tracheobronchial lymph nodes and tonsil) were collected for microscopic examination of lesions characteristic of PCV2 disease and PCV2 colonization via immunohistochemistry. In study 3, serum and body weights were collected for evaluation of *Mhyo-*specific antibodies and the impact of the vaccination in the body weight gain, respectively. At necropsy, lungs were scored for macroscopic lesions consistent with *Mhyo*-pneumonia and lung tissue was collected for pathological evaluation.

Treatments, housing, husbandry, and euthanasia methods in all groups were aligned with the European Union Guidelines and Good Clinical Practices. Animals were housed and managed following the guidelines depicted in the Guide for the Care and Use of Laboratory Animals of the National Institutes of Health. During the vaccination phase of study 1 all animals were randomly housed in one single room containing 6 different pens. Prior to challenge, animals were moved into a BSL-2 facility that consisted in 3 rooms with 2 independent pens per room (~10 animals per pen). During study 3, pigs were randomly allocated in 1 facility containing four independent pens. In all three studies animals were kept with the same pen mates throughout the study. Piglets within litters were kept together whenever possible throughout the study. Housing during study 2 mimicked that of study 1. All study protocols were approved by the Zoetis Institutional Animal Care and Use Committee, with the following reference numbers: studies 1 and 2, # KZ-3087e-2014-10-tkh; and study 3, 17-NARDO-04. Stocking density followed the Site SOPs, and the FASS Consortium Guide for Care and Use of Agricultural Animals in Research and Teaching, 3rd Edition, 2010.

### 2.2. Vaccination

Piglets were between 18 and 25 days-old at vaccination. One intramuscular dose of 2 mL of a trivalent vaccine containing PCV1—2a Chimera, PCV1—2b Chimera, both killed whole viruses, *Mhyo* bacterin, and 10% SP Oil adjuvant was administered into the right neck to T02, T04, and T07. Piglets from T01, T03, and T06 received one intramuscular dose of 2 mL into the right neck of a knock-out vaccine containing PCV2a and PCV2b or *Mhyo* and 10% SP Oil adjuvant, respectively. Piglets from T05 did not receive any vaccine.

### 2.3. Challenge

Twenty-three weeks post-vaccination all pigs were challenged. Piglets from study 1 were inoculated with a total of 4 mL of a PCV2a field strain (isolate 40895) containing approximately 10^6^ TCID_50_ per dose. Two mL were administered intramuscularly, and the remaining 2 mL were administered intranasally (1 mL per nostril). In study 2, all piglets were inoculated using the same procedures and route described for study 1 but using a PCV2b strain (isolate Fd7) containing approximately 10^6.2^ TCID_50_ per dose. In study 3, all piglets were inoculated with 20 mL of the *Mhyo* 232 strain diluted 1:1000 in Friis Medium and administered intratracheally. For the intratracheal challenge pigs were manually restrained by a qualified technician and a speculum was used to hold the mouth open. A catheter was manually placed into the trachea; once correct placement of the catheter was confirmed, the challenge material was injected via syringe followed by a small volume of air to clear the catheter. A separate catheter was used for each pig.

### 2.4. Clinical Observations 

Before vaccination all pigs were evaluated to ensure that only clinically healthy pigs were enrolled in the study. Around 1 h after vaccination and post challenge abnormal clinical signs all, such as depression, increased respiration, vomiting and lameness were evaluated in all animals. In addition, general health observations were done every day in the three studies.

### 2.5. PCV2 Real-Time Quantitative PCR 

A commercial kit (QiaAmp Blood 96 kit, Qiagen, Valencia, CA) was used to extract the DNA from serum and fecal swab samples collected on Day 0, 159, 164/165, 168, 171/172, 175, 178/179, and 181 in studies 1 and 2. A real-time quantitative PCR (qPCR) was used for the quantification of PCV2 genome. The program was done taking five microliter of DNA as template and consisted on a reverse transcription of 2 min at 50 °C, a denaturion phase of 10 min at 95 °C, and a total of 40 cycles, each cycle consisted of 25 s at 95 °C for denaturation and 1 min at 60 °C for annealing/extension. The process was performed using a thermocycler (Bio-Rad CFX 96). The following primers and probe sequences were used:Forward primer (P1591F), 5′ TGG CCC GCA GTA TTT GAT T 3′ (Final concentration 5 µM);Reverse primer (P1642R), 5′ CAG CTG GCA CAG CAG TTG AG 3′ (Final concentration 5 µM);Probe (P1591Probe), 5′ FAM-CCA GCA ATC/ZEN/AGA CCC CGT TGG AAT G-IABkFQ 3′ (Final concentration 10 µM).

All reactions with test samples were done in duplicate. In addition, a standard curve was generated by performing three replicates for each of the six progressive 1:10 dilutions of a seminal standard containing a known number of copies. Finally, to quantify the amount of virus, the quantity of DNA copies found on a 5 μL reaction were multiplied x 200 to obtain the number of copies/mL.

### 2.6. PCV2 Pathological Studies

At necropsy samples from tonsil and lymph nodes (inguinal, mesenteric, and tracheobronchial) were taken from all pigs involved in studies 1 and 2 and fixed by immersion in 10% buffered formalin. Later, samples were dehydrated and embedded in paraffin wax to ensure the preservation of tissue structure. Then, all samples were sent to the Iowa State University Veterinary Diagnostic Laboratory for histopathological analysis and for PCV2-antigen detection by immunohistochemistry (IHC). The histopathological examination was performed on 4 μm thick sections stained with hematoxylin and eosin. A blinded evaluation was performed by giving a subjective score for severity, from 0 (none) to 3 (severe), of lymphoid depletion (LD) and histiocytic replacement (HR). An animal was considered to have lesions if one or more tissues had a value over 0 and the final result was recorded as Yes (+) or No (-).

IHC was performed on the mentioned tissues by using a rabbit polyclonal antiserum against PCV2, following a previously defined protocol [25]. The quantity of PCV2 antigen found in the mentioned tissues were recorded as 0 (no staining) and 1–3 (different levels of staining). A score of 0 was considered negative while a score equal or higher to 1 was considered as positive.

### 2.7. Mhyo Pathological Studies

In study 3, half of the non-vaccinated control animals (T05) were necropsied approximately 12 weeks after vaccination. The remaining non-vaccinated control pigs were necropsied on the challenge day, prior to challenge. Four weeks after challenge all pigs were necropsied. Each pig was anesthetized, exsanguinated and necropsied. At necropsy *Mhyo* macroscopic lesions in the lungs were evaluated on a range from 0% to 100% for consolidation for each lobe (left cranial, left middle, left caudal, right cranial, right middle, right caudal, and accessory) by a veterinarian. A representative section of lung tissue was dissected from each animal, fixed in 10% buffered formalin solution and evaluated for positive or negative *Mhyo* status via IHC at the UMN-VDL. A labeled streptavidin–biotin detection kit (DAKO) was used on paraffin-embedded tissue sections to detect the presence of *Mhyo.* Before applying the *Mhyo* monoclonal antibody (identification number D79DI–7; Richard Ross, Iowa State University) samples were dewaxed and rehydrated, covered with 1:10 ethylenediaminetetraacetic acid butter solution, pH 6.0 (Richard Allan Scientific, Kalamazoo, MI, USA), boiled for 5 min and after cooling for 20 min, the slides were rinsed. Then the mentioned antibody was applied for 2 h at room temperature in a 1:500 dilution. Later, the detection kit (DAKO) was used. Stained slides were scored on the following way: 0, no signal detectable; 1, weak labeling lining the ciliated epithelium of at least one airway; 2, weak-to-moderate labeling on the surface of a low number of airways and 3, intense labeling on the surface of several airways.

### 2.8. Serology

The existence of antibodies against PCV2 in serum samples obtained throughout studies 1 and 2 was evaluated using an indirect ELISA test. Briefly, positive capture antigen (Sf9 cells infected with recombinant baculovirus expressing PCV2 capsid protein) was used to coat 96-well polystyrene plates. Some other wells were coated with conventional Sf9 cells to act as a negative control. Later, the plates were treated with blocking reagent and incubated with sera from test samples. A sample know to be positive for PCV2 antibodies was added to wells containing positive capture antigen as well as to wells coated with negative capture antigen to act as a positive control. The secondary antibody, a goat anti-swine antibody conjugated with HRP, was then added. Finally, the peroxidase substrate (TMB) was pipetted and incubated for 20 ± 2 min. The color generated was quantified by an ELISA plate reader. Each reagent added to the immuno-plates was incubated according to protocol and was ringed to remove the excess reagent before to each step. The OD value of positive control and test samples was calculated by subtracting average OD of negative control from average OD of test samples and positive control. Serum antibody value was expressed as S/P (sample/positive control) ratio, which is the OD of the test sample divided by that of the positive control sample.

Serum samples obtained from study 3 were analyzed by the University of Minnesota, Veterinary Diagnostic Laboratory (UMN-VDL). An ELISA blocking assay (Dako Corporation, Carpenteria, CA, USA) was used to detect the existence of *Mhyo* antibodies. Briefly, *Mhyo* antigen was used to coat the 96-well plates on which serum samples were incubated. Peroxidase conjugated mouse monoclonal antibody to *Mhyo* was pipetted to the plates. The conjugated antibody competed with any *Mhyo* antibody present in the sample serum for binding siters on the immobilized antigen. After 15 min of incubation, wells were ringed and the substrate was added. The color reaction was halted by the action of the acid. The intensity of color in measured at 450 nm and compared with the absorbance of buffer controls. Samples with a mean OD value lower than 50% of the OD obtained from buffer control were considered as positive.

### 2.9. Statistical Analysis

A centralized data management system (SAS/STAT User’s Version 9.4, or higher, SAS Institute, Cary, NC, USA) was used to perform all data summaries and analyses. An appropriate logarithm transformation was applied to the results before performing the statistical analysis, if necessary.

Frequency distribution of PCV2 viremia, PCV2 fecal shedding, microscopic lesions, and IHC scores were calculated per treatment and time point data were collected. Animals that were viremic or shed at any time in the study (ever viremic or ever shed) were analyzed with a generalized linear repeated measures mixed model with fixed effect treatment and random effects room, pen with room, and block within pen and room. If the treatment main effect was significant (*p* ≤ 0.05), pair-wise treatment comparisons were made. If the mixed model did not converge, Fisher’s Exact test was used for analysis.

The stratified prevented fraction was used to summarize if the animal ever shed PCV2 or had any abnormal LD, HR, or IHC results with 95% confidence intervals.

Viremia, viral load in feces, and PCV2 serology data were analyzed with a generalized linear repeated measures mixed model with fixed effect treatment, time point, and treatment by time point interaction and random effects room, pen with room and block within pen and room, and animal within pen, room, block, and treatment (which is the animal term). Comparisons between treatments were made at each time point. A 5% level of significance (*p* ≤ 0.05) was applied to evaluate the statistical differences. Least squares means (back transformed for viremia, viral load in fecal shedding and serology), standard errors, 95% confidence intervals of means and ranges were calculated per treatment and time point. Serology means prior to vaccination (Day 0 or Day −1) were not model based.

Percentage of total lung lesions consistent with *Mhyo* infections was estimated using the subsequent method: Percentage of total lung with lesions = {(0.10 × left cranial) + (0.10 × left middle) + (0.25 × left caudal) + (0.10 × right cranial) + (0.10 × right middle) + (0.25 × right caudal) + (0.10 × accessory)} [26]. The arcsine square root transformation was applied to the percentage of total lung with lesions before the analysis. A general linear mixed model with fixed term treatment random terms: pen, and block within pen was used to calculate the percentage of Total Lung with Lesions. When a significant treatment effect was found, pair-wise comparisons were made between treatment groups. Back transformed least squares means of percentage of total lung with lesions, and their 95% confidence intervals were calculated as well as the minimums and maximums. 

Frequency distributions were calculated for each treatment groups considering the next: 0% to <5%, 5% to <10%, 10% to <20%, 20% to <30%, 30% and greater.

*Mhyo* serology results were log transformed prior to analysis and a general linear mixed model with repeated measured with fixed effects: treatment, time point and treatment by time point interaction and random effects: pen, block within pen, and animal within block, pen, treatment, which is the animal term, was used for the analysis. After testing for a significant (*p* ≤ 0.05) treatment effect or treatment by time-point interaction linear combinations of the parameter estimates were used in a priori contrasts. 

Body weights were examined using a general linear repeated measures mixed model with fixed effects: treatment, time point, and treatment and time point interaction and random effects: pen, block within pen, and animal within pen, block, and treatment (which is the animal term). Treatment least squares means, 95% confidence intervals, the minimum and maximum were calculated for each time point data are collected. If the treatment effect or treatment by time point interaction was significant, pair-wise treatment comparisons were made between groups.

## 3. Results

### 3.1. PCV2 Viremia

The results of PCV2 viremia observed in studies 1 and 2 are summarized in Figure 1 and Table 2. All pigs were negative to PCV2 presence in blood before challenge (Day 159). On study 1, PCV2a viremia was initially detected four days after challenge (Day 164) in control group only. From Day 168 through Day 175, mean viremia in the vaccinated pigs (T02) was significantly inferior (*p* ≤ 0.0011) than that of the control group (T01). The percentage of pigs that showed viremia at any point of the study was also significantly lower (*p* = 0.0001) in vaccinated group (T02) to that in control pigs (T01). A clinically relevant prevented fraction, with a lower 95% interval >0, was also demonstrated.

On study 2, PCV2b viremia was initially detected five days after challenge (Day 165). Viremia was significantly lower (*p* ≤ 0.0029) in the vaccinated group (T04) from Day 168 until study completion (Day 181), when compared to controls (T03). The percentage of pigs with viremia at any time in the study was significantly inferior (*p* = 0.0056) in vaccinated group (T04) than in control group+ (T03). Like in study 1, a clinically relevant prevented fraction, with a lower 95% interval >0, was also demonstrated in Study 2.

### 3.2. PCV2 Fecal Shedding

The results of PCV2 fecal shedding detected in studies 1 and 2 are summarized in Figure 2 and Table 3. Before challenge (Day 159) none of the pigs shed PCV2 through the fecal route. In study 1, PCV2a shedding was initially detected four days after challenge (Day 164) and continued until the study completion day (Day 181). From Day 171 through Day 178, mean fecal shedding was significantly lower (*p* ≤ 0.0002) in vaccinated pigs (T02) when compared to controls (T01). The percentage of positive pigs was also significantly lower (*p* = 0.0102) in vaccinated pigs (T02). A clinically relevant prevented fraction (lower 95% interval >0) was also demonstrated. Interestingly on Day 164 vaccinated pigs showed a numerically higher number of copies when compared to the control group. This peak of shedding right after challenge may have explained by the time taken to mount a robust immune response. This response was successfully assembled, and later vaccinated animals showed lower shedding when compared to controls.

In study 2, PCV2 shedding was initially detected five days after challenge (Day 165) and was detected on each timepoint until study completion in control group (T03). Shedding was only detected on Day 165 and Day 179 in vaccinated animals (T04). From Day 172 until Day 181, mean fecal shedding was significantly inferior (*p* < 0.0001) in vaccinates (T04). The percentage of ever positive pigs was significantly less (*p* < 0.0001) in the vaccinated pigs (T04) and a clinically relevant prevented fraction (lower 95% interval >0) was also demonstrated.

### 3.3. PCV2 Pathological Findings

Table 4 shows the number of pigs with a positive score on histopathological results for studies 1 and 2. In study 1, the percentage of pigs positive for IHC did not show significant differences (*p* = 0.0515) between vaccinated (T02) and control pigs (T01). However, the lower 95% confidence interval of the prevented fraction was >0 (0.2144). The lack of statistical significance in these measures was likely due to the low number of animals positive for PCV2 staining of the lymph node (five in T01 and zero in T02). Differences were also not significant in the percentage of positive pigs among groups for LD (*p* = 0.9560) or HR (*p* = 0.4808). Regarding the severity of the lesions all lesions described had a score of 1 with the only exception of one control animal that showed a score of 2 on LD in the mesenteric lymph node.

In study 2, the percentage of pigs positive for IHC were significantly lower (*p* = 0.0098) the vaccinated pigs (T04) when compared to controls (T03). The lower 95% confidence interval of the prevented fraction was >0 (0.4256). In addition, the percentage of positive pigs for HR was significantly different based on vaccination status (*p* = 0.0306). On the other hand, differences were not significant for the percentage of pigs positive for LD (*p* = 0.0544). The lower 95% confidence interval of the prevented fraction was <0 for both variables (−0.022471 for HR and −0.087594 for LD). Regarding the severity of the lesions one vaccinated animal had a score of 2 on HR in the tracheal lymph node, while four control animals showed repeated scores of 2 mainly on HR and LD in most lymph nodes tested.

### 3.4. Mhyo Macroscopic Lung Lesions

Table 5 shows the frequency distributions of total macroscopic lung lesion categories by treatment. Animals belonging to the non-vaccinated control group (T05) did not show any macroscopic lung lesions, indicating that those animals remained free of *Mhyo* (or other disease which have confounded the interpretation of *Mhyo* challenge outcome) throughout the pre-challenge study period. At necropsy, the average percentage of lung presenting gross lesions was significantly superior (*p* = 0.0196) in vaccine control animals (T06) when compared to vaccinated (T07) (Table 6). In the vaccine control group (T06) average percentage of lung with lesions was 8.2 while in the vaccinated group (T07) was 4.3.

### 3.5. Serology

In both studies 1 and 2, the mean PCV2 antibodies detected by ELISA declined from vaccination (Day 0) until the day before challenge (Day 159), Figure 3. This drop in antibody levels was more acute in control groups (T01 and T03) compared to vaccinates and was significantly lower when compared to vaccinated pigs (T02 and T04) on Day 159 (*p* = 0.0005 on study 1 and *p* < 0.0001 on study 2, respectively). From Day 159 until study completion, antibody levels increased in all treatment groups in both studies. However, vaccinated animals (T02 and T04) continued having significantly higher antibody values *p* < 0.0001 in study 1 and *p* < 0.0001 in study 2, respectively, when compared to controls (T01 and T03). 

All pigs in study 3 were negative for *Mhyo*-specific antibodies by a competitive ELISA prior to vaccination (Table 7). Prior to challenge (Day 159), all non-vaccinated control group (T05) and vaccine control group (T06) pigs remained *Mhyo* antibody negative. Before challenge (Day 159) and at necropsy (Day 187/188), the vaccinated group (T07) had significantly higher ELISA levels (*p* ≤ 0.0068) in comparison to the vaccine control group (T06). 

### 3.6. Body Weights

Body weights were collected for vaccine control (T06) and vaccinated (T07) groups in study 3. Differences in mean weights were not significant (*p* > 0.05) between any of the treatment groups was described on any time (Table 8).

### 3.7. Clinical Signs

In study 1, five control (T01) and seven vaccinated animals (T02) showed clinical signs after challenge. Those clinical signs were in all cases related with problems in claws or limbs. Regarding the adverse events one animal from the control group (T01) was found death prior to challenge on day 109 of study. Necropsy showed diffuse pericarditis, peritonitis, mild pleuritis, and less than 10% lung consolidation. Bacterial analysis was positive for mixed bacteria and *E. coli* from the peritoneum and pericardium. Six additional animals were withdrawn from the study after challenge, between Days 161 and 175 (five from the vaccinated group (T02) and one from the control group (T01), due to lameness second to abraded or torn claws. Those six animals were omitted from the tissue analysis and three out of those six were omitted from PCV2 viremia and shedding analysis as they were treated with antibiotics during the study. Three additional animals treated with antibiotics during the study were omitted from viremia and shedding analysis. 

In study 2, two control (T03) and three vaccinated animals (T04) showed clinical signs after challenge. Those clinical signs were in all cases related with problems in claws or limbs. Regarding the adverse events one animal from the control group (T03) was found death prior to challenge on day 91 of study. Necropsy showed edematous hemorrhagic lungs. Seven animals (four belonging to the vaccinated group (T04) and three controls (T03)) were withdrawn from the study, three before challenge and the remaining four after challenge, between Days 162 and 179 due to lameness second to abraded or torn claws. Those seven animals were omitted from tissue analysis and four were also omitted from PCV2 viremia and shedding analysis as they were treated with antibiotics during the study. 

In study 3, one animal from the non-treated control group (T05), one animal from the vaccinated control (T06), and three animals from the vaccinated group (T07) were found dead prior to challenge. Necropsies suggested meningitis due to *Strep. suis* in the T05 pig; severe peritonitis, arthritis in one leg and torsion of elongated mesentery in the three pigs from T07. The T06 animal that died was found stuck between the fence and the waterer. In addition, one animal from the vaccinated group (T07) died after challenge. Necropsy revealed presence of fluid in the abdomen and thoracic cavity; bacterial culture from the fluid sample resulted in isolation of *H. parasuis*.

## 4. Discussion

The goal of this study was to assess the efficacy of a novel ready-to-use vaccine including PCV2a, PCV2b, and *Mhyo,* after one dose of intramuscular administration to pigs at 21 ± 4 days of age, against a pathogenic challenge 23 weeks after vaccination. With that aim, three independent studies were performed, each study evaluating one of the fractions included in the vaccine. Vaccination with PCVa-PCV2b-*Mhyo* significantly protected animals from PCV2 associated disease as well as respiratory disease caused by *Mhyo*.

PCV2 and *Mhyo* are both critical respiratory pathogens in swine. Piglet vaccination around three weeks of age, corresponding with weaning and pig handling, is one of the main strategies to control both pathogens during the nursery and fattening periods [20]. Combined application of PCV2 and *Mhyo* vaccines suppose an advantage for farmers and veterinarians in terms of economizing pig handling events and associated management/labor costs. Importantly, administration of a multi-valent PCV2-*Mhyo* vaccination is beneficial for animal welfare. The first commercial approach to combine both PCV2 and *Mhyo* vaccines in one dose consisted in the combination of two already licensed products from the same manufacturer [27]. Soon, this ready-to-mix vaccine evolved to the development and commercialization of ready-to-use products [28]. In addition to the benefits derived from the combined multivalent application, the ability of vaccines to confer durable and stable protection is a key asset. In intensive pig production the average fattening time is around six months. Thus, vaccines with a DOI close to this time period suppose a clear advantage as they are able to confer protection during the entire fattening period.

This vaccine includes both a PCV2a and a PCV2b fraction in a single bottle. Most PCV2 vaccines in the market are based on PCV2a technology only and all (with the exception of the current formulation) include a single genotype. PCV2a based vaccines have been a success story for the swine industry and the vast majority of pigs are vaccinated against PCV2 worldwide. Despite vaccine use and the ensuing benefit to swine in terms of reduced occurrence and severity in clinical signs, PCV2 seroprevalence continues with some herds still reporting clinical PCV2. In addition to vaccine failures associated with clinical disease, subclinical PCV2 infection remains a major problem for economic loss even for herds that vaccinate [29].

While PCV2a vaccines have been demonstrated to generate some cross-protection from clinical PCV2, associated disease PCV2 viruses continue to evolve and instances of vaccine failures due to diverse PCV2 field virus challenge have been documented [30]. PCV2 vaccines generally do not provide sterilizing immunity [30,31] and historic single genotype PCV2 vaccines may not induce broad enough cross-protection to cover evolving field viruses [32]. In fact, traditional single genotype PCV2 vaccine use has contributed to the selection of emergent viruses [33]. Emergent field viruses are especially different from vaccine strains in capsid epitopic regions [30,31]. Since the capsid is the immunodominant protein, shifts away from vaccine capsid epitopes may be contributing the vaccine failures. It is noteworthy that traditional PCV2a vaccines have clearly added to the selection of PCV2a field viruses and less to the selection and diversity of PCV2b and PCV2d viruses [33,34,35,36]. This may reasonably imply that PCV2 vaccine induced coverage is best when the vaccine virus and challenge virus are closely matched. 

PCV2b and PCV2d have surpassed PCV2a in terms of prevalence in many geographies and new PCV2 genotypes including those with recombinant virus origins are continually discovered [6]. Importantly, all three, PCV2a, PCV2b, and PCV2d, continue to circulate. Studies of PCV2 phylogeny suggest two major groups of PCV2, namely, the PCV2a and PCV2b/d genogroups [37]. This is reasonable as PCV2b and 2d share more common epitopes with each other than PCV2a shares with PCV2d [26,38,39]. Taken together, having a vaccine which contains both PCV2a and PCV2b, increases the total number of epitopes in the vaccine. Thus, the PCV2a-PCV2b vaccine has the potential to educate the pig’s immune system to diverse PCV2 viruses including these three disease-causing circulation strains. 

In most markets around the world, pigs are slaughtered around 26 weeks of age. Considering that the vaccine evaluated here was administered at three weeks of age, the challenge date, 23 weeks (Day 159), was selected to ensure that the vaccine confers protection during the entire fattening period. Other swine vaccines available in the market target similar duration of immunity period when dosed at three weeks. However, none of them reached a duration of immunity longer than 23 weeks for both the PCV2 and the *Mhyo* fractions, when administered, combined and as a single dose to three weeks old pigs [40,41,42,43,44]

Protective efficacy in studies 1 and 2 is evidenced by fewer percent of animals ever viremic, and fewer percent of animals ever fecal shedding following a challenge. Although fewer percent of animals were positive for IHC, LD, or HR; differences were not significative. This may have been explained by the low amount of pigs with pathological changes and abnormal clinical signs after challenge in both groups. Similar PCV2 subclinical infection outcome has been previously described in PCV2 experimental inoculations which is consequent to that described in previous PCV2 experimental infections [24,45]. Finally, criteria for prevention/reduction in viremia, fecal shedding, and PCV2 antigen in lymphoid tissues (IHC) were met as those variables presented a value >0 on the lower 95% confidence interval for the prevented fraction.

Study 3 was also considered as valid as animals did not have pre-challenge exposure to *Mhyo.* The experimental model used for infection was based on that described in previous studies including the same challenge strain [24,40]. This challenge demonstrated to be valid as the back-transformed mean lung lesion scores observed in control pigs (T06) was greater than 3%. Vaccination significantly protected animals from *Mhyo* infection as evidenced by the significantly lower percentage of lung lesions detected in vaccinated pigs (T07) following challenge. In addition, vaccinated pigs (T07) showed significantly higher level of antibodies at necropsy, when compared to control pigs (T06). Although vaccination did not have any impact in the body weights of the animals it can be concluded that vaccination was efficacious in building up a stronger antibody response and preventing lung lesions at necropsy.

## 5. Conclusions

The efficacy of one dose of the three fractions included in the vaccine (PCV2a/PCV2b/*Mhyo*) against a virulent challenge at 23 weeks was verified by an enhancement in virological/bacteriological, immunological, and clinical variables. Therefore, a DOI of at least 23 weeks was demonstrated for each of the three components of the trivalent vaccine tested.

## Figures and Tables

**Figure 1 vaccines-09-00834-f001:**
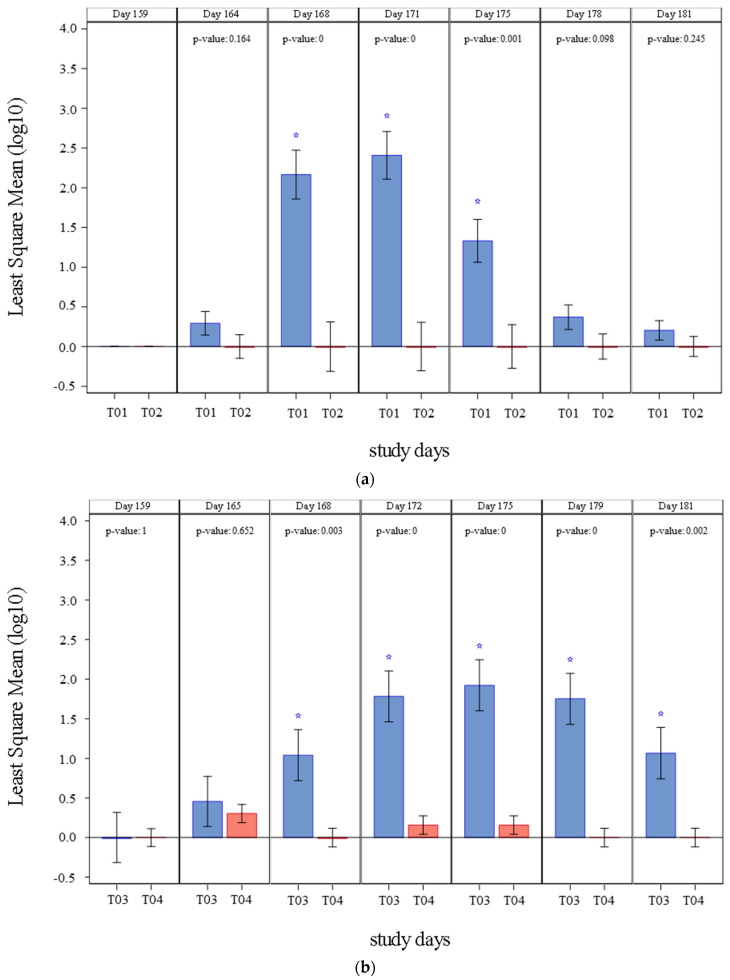
PCV2 viremia (Least Square Mean (log_10_) of DNA copies/mL) by treatment and day of study and significance: (**a**) PCV2 viremia detected in study 1, results from T01 are represented in blue and results from T02 are represented in red. When *p*-value < 0.0001 it is described as 0, and significant differences between groups (*p* ≤ 0.05) are marked with a blue star; (**b**) PCV2 viremia detected in study 2, results from T03 are represented in blue and results from T04 are represented in red. When *p*-value < 0.0001 it is described as 0, and significant differences between groups (*p* ≤ 0.05) are marked with a blue star.

**Figure 2 vaccines-09-00834-f002:**
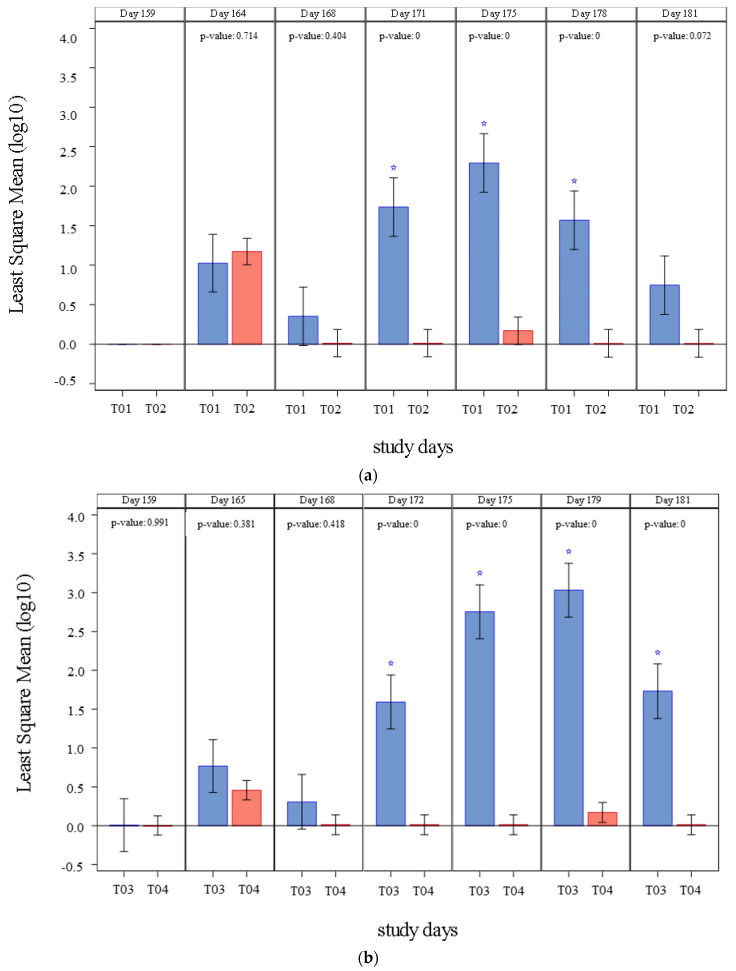
PCV2 fecal shedding (Least Square Mean (log_10_) of DNA copies/mL) by treatment and day of study and significance: (**a**) PCV2 fecal shedding detected in study 1, results from T01 are represented in blue and results from T02 are represented in red. When *p*-value < 0.0001 it is described as 0, and significant differences between groups (*p* ≤ 0.05) are marked with a blue star; (**b**) PCV2 fecal shedding detected in study 2, results from T03 are represented in blue and results from T04 are represented in red. When *p*-value < 0.0001 it is described as 0, and significant differences between groups (*p* ≤ 0.05) are marked with a blue star.

**Figure 3 vaccines-09-00834-f003:**
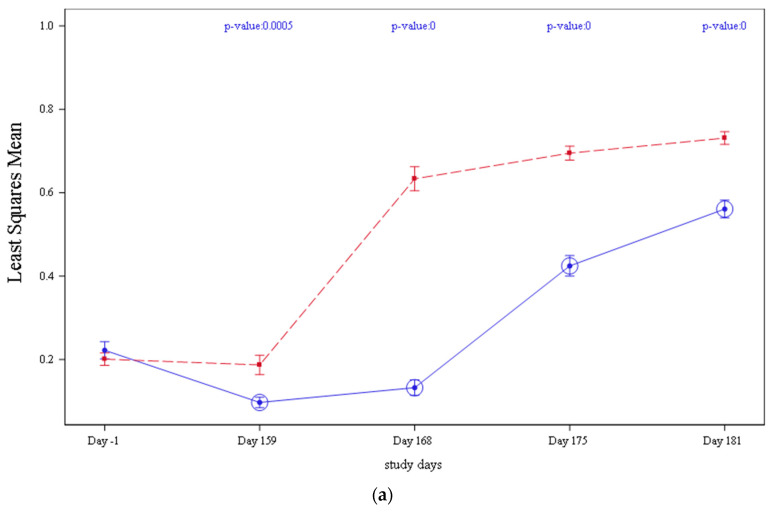
PCV2 serology (Least Squares Mean) by treatment and day of study and significance values: (**a**) PCV2 serology detected on study 1, results from T01 are represented in blue and results T02. are represented in red, when *p*-value < 0.0001 it was represented as 0; (**b**) PCV2 serology detected on study 2, results from T03 are represented in blue and results from T04 are represented in red, when *p*-value < 0.0001 it was represented as 0.

**Table 1 vaccines-09-00834-t001:** Independent studies, treatment groups, and vaccination and challenge patterns involved in this experiment.

Study Number	Treatment	Number of Animals	Vaccine	Challenge
Study 1	T01	30	*Mhyo* fraction	PCV2a
	T02	31	PCV2a, PCV2b and *Mhyo* fractions	
Study 2	T03	30	*Mhyo* fraction	PCV2b
	T04	30	PCV2a, PCV2b and *Mhyo* fractions	
Study 3	T05	10	Not vaccinated	Not challenged
	T06	71	PCV2a and PCV2b fractions	*Mhyo*
	T07	71	PCV2a, PCV2b and *Mhyo* fractions	

**Table 2 vaccines-09-00834-t002:** Percent of ever positive pigs and estimate of prevented fraction for PCV2 viremia.

Study	Treatment	Number (%) of q PCR Positive Pigs	*p*-Value	Prevented Fraction		
				Vs T01	Lower 95% Bound	Upper 95% Bound
Study 1 (PCV2a)	Control (T01)	17/28 (60.7)	0.0001	1.0000	0.7926	1.0000
	Vaccinated (T02)	0/25 (0)				
Study 2 (PCV2b)	Control (T03)	15/27 (55.6)	0.0056	0.72388	0.28022	0.89408
	Vaccinated (T04)	4/26 (15.4)				

**Table 3 vaccines-09-00834-t003:** Percent of ever positive pigs and estimate of prevented fraction for PCV2 fecal shedding.

Study	Treatment	Number (%) of q PCR Positive Pigs	*p*-Value	Prevented Fraction		
				Vs Control	Lower 95% Bound	Upper 95% Bound
Study 1 (PCV2a)	Control (T01)	18/27 (66.7%)	0.0102	0.56495	0.15111	0.77703
	Vaccinated (T02)	8/27 (29.6%)				
Study 2 (PCV2b)	Control (T03)	21/26 (80.8%)	<0.0001	0.84944	0.58637	0.94520
	Vaccinated (T04)	4/27 (14.8%)				

**Table 4 vaccines-09-00834-t004:** Results of the histopathological analysis of the different studied lymphoid tissues from vaccinated and control pigs after PCV2 challenge.

			Number of Pigs with a Score > 0/Number of Pigs Tested *	
Study	Tissue	Parameter	Control (T01)	Vaccinated (T02)
Study 1 (PCV2a)	Mesenteric Lymph Node	IHC **	1/27	0/25
		LD ***	1/27	1/25
		HR ****	0/27	1/25
	Superficial Inguinal Lymph Node	IHC	4/27	0/25
		LD	0/27	0/25
		HR	0/27	0/25
	Tonsil	IHC	2/27	0/25
		LD	0/27	0/25
		HR	0/27	0/25
	Tracheobronchial Lymph Node	IHC	0/27	0/25
		LD	0/27	0/25
		HR	0/27	0/25
			Control (T03)	Vaccinated (T04)
Study 2 (PCV2b)	Mesenteric Lymph Node	IHC	4/25	0/26
		LD	6/25	2/26
		HR	8/25	2/26
	Superficial Inguinal Lymph Node	IHC	1/25	0/26
		LD	6/25	2/26
		HR	6/25	2/26
	Tonsil	IHC	4/25	0/26
		LD	5/25	0/26
		HR	3/25	0/26
	Tracheobronchial Lymph Node	IHC	4/25	0/26
		LD	6/25	1/26
		HR	6/25	1/26

* An animal was considered positive if score >0. ** IHC: Immunohistochemistry. *** LD: Lymphoid Depletion. **** HR: Histiocytic Replacement.

**Table 5 vaccines-09-00834-t005:** Frequency distributions of total *Mhyo*-like macroscopic lung lesion categories by treatment.

Treatment	Lung Lesion Category										Total Observations
	0% ≤ to <5%		5% ≤ to <10%		10% ≤ to <20%		20% ≤ to <30%		30% ≤ %		
	Number	%	Number	%	Number	%	Number	%	Number	%	Number
Non-vaccinated control (T05)	9	100.0	0	0.0	0	0.0	0	0.0	0	0.0	9
Control (T06)	26	37.1	10	14.3	18	25.7	10	14.3	6	8.6	70
Vaccinated (T07)	37	55.2	10	14.9	9	13.4	8	11.9	3	4.5	67

**Table 6 vaccines-09-00834-t006:** Least squares means and significance values for *Mhyo*-consistent macroscopic lung lesions.

Treatment	Summary of Least Squares Means ^1^				Contrast vs. T06
	Number of Animals	Mean % Lung with Lesions	Std Deviaton/Std Error ^2^ % Lung with Lesions	Range % Lung with Lesions	*p*-Value
Non-vaccinated control (T05)	5 (Day 84)	0.0	0.00	0.0 to 0.0	Na ^3^
	4 (Day 159)	0.0	0.00	0.0 to 0.0	Na
Control (T06)	70	8.2	1.45	0.0 to 45.0	Na
Vaccinated (T07)	67	4.3	1.09	0.0 to 35.0	0.0196

^1^ Back Transformed Least Squares; ^2^ Standard Deviation for T05, Standard Error for T06 and T07; ^3^ Not available.

**Table 7 vaccines-09-00834-t007:** *Mhyo* serology (Least Squares Means, Standard Errors and Significance values) per treatment and day of study.

Treatment		Day −1	Day 159	Day 187/188
Control (T06)	Geometric Least Squares Mean	98.7941	78.0069	51.8656
	Number of observations	71	70	70
	Standard Errors	1.5117	1.4927	3.1159
Vaccinated (T07)	Geometric Least Squares Mean	98.8315	72.9733	27.7770
	Number of observations	71	68	67
	Standard Errors	1.5434	1.7582	3.9046
Significance vs. T06		*p* = 0.9735	*p* = 0.0068	*p* ≤ 0.0001

**Table 8 vaccines-09-00834-t008:** Least squares means and significance values for body weights.

Treatment	Least Squares Mean (kg)		
	Day −1	Day 159	Day 187/188
Control (T06)	5.1	116.6	130.4
Vaccinated (T07)	5.2	116.0	128.4
Significance vs. T06	*p* = 0.6796	*p* = 0.8298	*p* = 0.4945

## Data Availability

Data is contained within the article.

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
