# Peer review of "One Dose of a Novel Vaccine Containing Two Genotypes of Porcine Circovirus (PCV2a and PCV2b) and *Mycoplasma hyopneumoniae* Conferred a Duration of Immunity of 23 Weeks"

_vaccines, 2021, doi:10.3390/vaccines9080834_

Round 1
Reviewer 1 Report
This manuscript describes a single dose vaccine containing two genotypes of porcine circovirus (PCV2) as well as Mycoplasma hyopneumoniae bacterin. These pathogens play a role in Porcine Respiratory Disease Complex. Currently, separate vaccines are available for one genotype of PCV2 and Mhyo. The current PCV2 vaccine targets genotype PCV2a but PCV2b/2d have become the predominant genotypes in commercial swine populations. The goal of this study was to improve PCV2 vaccine but targeting both PCV2a and PCV2b in a single vaccine while also pairing this with Mhyo vaccination. If a multi-valent vaccine strategy proves successful, it has potentially positive economic and animal welfare benefits. The study was well-designed and assessed vaccinated animals following challenge with each of the three pathogens targeted by the vaccine in separate studies. The manuscript presents the work in an easy to follow manner, but some additional clarification (detailed below) will be helpful in evaluating whether the conclusions are supported by the results.
Specific comments
Introduction – Nicely written introduction. A few additional lines as to what are known or suspected transmission routes for these pathogens would be helpful to set the stage for whether certain sample types (e.g. fecal samples) are relevant to transmission.
Methods
- A table summarizing the vaccination and challenge groups would be very helpful for quick reference. Such a table might include group, study number, number of animals, vaccine, and challenge
- Please clarify how groups were housed. Were all the animals in study 1 housed as a single, mixed group or were the animals receiving the Mhyo only vaccine housed separately from those receiving the combined vaccine? What about the animals from studies 2 or 3?
- How was the equation for viral load using RT-qPCR results determined? Please add a reference if available. If no published reference is available, please add detail on how this equation was determined (e.g. was it based on a standard curve? If so, what was the DNA template for the standard curve?)
- It is unclear what the authors mean by ‘ever’ viremic and ‘ever’ shed. Please consider using an alternative such as ‘always’ if that is what is meant by ‘ever’. This also applies to the results and discussion sections.
Results
- There are two Table 2 in the results section. Please correct the first one to be Table 1.
- The % positive of T02 was 0 (presumed table 1) but figure 1a shows error bars (and possibly a small red bar) for T02 at all post-challenge time points and it is stated that mean viremia of the vaccinated group was significantly less than in the control group. Please clarify the text and table title to better reflect if the positive observation in T02 animals was observed in different animals at each of the time point.
- Section 3.3 and 3.4 – is it possible to add representative images of the pathological findings (for PCV2) and macroscopic lung lesions (Mhyo)? This help the reader understand the scoring that was applied. For PCV2, how often were pigs found to have positive scores across the different tissues assessed? When positive, did the vaccinated pigs tend to have lower scores? In table 3, it would be helpful to include what the LD and DR abbreviations stand for (maybe as a footnote) so the reader does not have to return to the methods text.
- It may be helpful to remind the reader that a competitive ELISA was used for Mhyo serology (section 3.5). This would help make the numerical values in table 6 make more sense (lower geo LSM demonstrates that there was more antibody present in the serum.
- Clinical signs are listed in methods section 2.4 yet observations of these signs were not presented in results section 3.7 (instead, section 3.7 seems to present intercurrent issues that were present in some of the animals). Please describe the clinical signs observed post-challenge in the vaccinated and control groups. A table of this information may also be helpful.
Discussion
- How do the inoculation amounts compare to what might be acquired naturally? Could this have impacted prevalence of pathological findings and clinical disease?
- Data clearly show that PCV2 is reduced in vaccinated animals but there appeared to be more positive pigs in the vaccinated group from study 1 when fecal shedding was assessed. Please add a comment as to what could be contributing to this observation.
- How does the protective efficacy observed in this study compare to currently available vaccines?
Author Response
Dear reviewer,
Thanks for taking the time to read and review this manuscript. A response to your questions can be found in the attachment.
Kind regards,

Reviewer 2 Report
The Editor Vaccine
Thank you for the opportunity to review the manuscript: “One dose of a novel vaccine containing two genotypes of por-2 cine circovirus (PCV2a and PCV2b) and Mycoplasma hyopneu-3 moniae conferred a duration of immunity of 23 weeks”.
The paper has been carefully reviewed but significant concerns arose:
Major Issue:
In M&M, which serological and DNA detection techniques were performed to evaluate negative Pigs (Line 97)?
For the serological evaluation and DNA detection, what was the frequency of blood and swab collection?
What was the reason for choosing this period (159 days) for the challenge?
To facilitate understanding, make a table with the experimental groups.
Table 2 - Abbreviations caption missing
Improve discussion, especially Mycoplasma results
Author Response
Dear reviewer,
Thanks for taking the time to read and correct our manuscript. In the attached file you will find the answer to your questions.
Kind regards,

Reviewer 3 Report
Gracia et al. (ID#: vaccines-1279204) assessed a trivalent vaccine against several of the key causative agents Porcine Respiratory Disease Complex (PRDC) in pigs. The aim was to determine duration of immunity against two genotypes of Porcine Circovirus, type 2a (PCV2a) and 2b (PCV2b), in addition to Mycoplasma hyponeumoniae (Mhyo), in separate challenge studies using each infectious agent independently. They also assessed immunogenicity and clinical parameters in challenge animals. Piglets were challenged with the various agents at 23 weeks post-infection. Animals were euthanized 3-4 weeks post-challenge. Results demonstrate that vaccinated piglets had reduced PCV2 viremia and viral shedding in feces. They found no differences in measured immunological or clinical parameters following PCV2a challenge but found differences in PCV2b and Myho-challenged animals. The results demonstrate that a combination vaccine protects from all three infectious agents.
Overall, the manuscript is well written, and the results are clearly described. The statistical analyses are appropriate, and all controls are included and clearly defined. The overall study design is excellent. The major conclusions are supported by the results. The results demonstrating the efficacy of a trivalent vaccine will be of interest to veterinarians and pork production farms. I have two major concerns and one minor concern, listed below. Once the authors have addressed these concerns, my recommendation is for acceptance for publication.
Major Concerns:
- The authors do not include any details on animal housing or veterinary medical care. Was this work overseen by a veterinarian? Was the study approved by an IACUC? A paragraph in the Methods section describing animal welfare should be included.
- A figure demonstrating sequence divergence between PCV2a and PCVb might help strengthen the argument for including the two specific PCV genotypes in this trivalent vaccine and might increase the impact of the research. Or alternatively, adding this information to the relevant Introduction paragraph, or adding a paragraph in the Discussion could suffice.
Minor Concern:
- Legend for Figure 3, Line 395: Is “Figure 02.” supposed to be “T02”?
Author Response
Dear reviewer,
Thanks for taking the time to read and correct our manuscript. In the attachment you will find the answer to your questions.
Kind regards,

Round 2
Reviewer 2 Report
Accept